# Hardware Resilience Properties of Text-Guided Image Classifiers

**Syed Talal Wasim**
Mohamed bin Zayed University of AI

**Kabila Haile Soboka**
Independent

**Abdulrahman Mahmoud**
Harvard University

**Salman Khan**
Mohamed bin Zayed University of AI

**David Brooks**
Harvard University

**Gu-Yeon Wei**
Harvard University

## Abstract

This paper presents a novel method to enhance the reliability of image classification models during deployment in the face of transient hardware errors. By utilizing enriched text embeddings derived from GPT-3 with question prompts per class and CLIP pretrained text encoder, we investigate their impact as an initialization for the classification layer. Our approach achieves a remarkable $5.5\times$ average increase in hardware reliability (and up to $14\times$) across various architectures in the most critical layer, with minimal accuracy drop ($0.3\%$ on average) compared to baseline PyTorch models. Furthermore, our method seamlessly integrates with any image classification backbone, showcases results across various network architectures, decreases parameter and FLOPs overhead, and follows a consistent training recipe. This research offers a practical and efficient solution to bolster the robustness of image classification models against hardware failures, with potential implications for future studies in this domain. Our code and models are released at https://github.com/TalalWasim/TextGuidedResilience.

## 1 Introduction

As transistors in hardware shrink in size, they become more susceptible to random bit-flips from environmental factors such as cosmic particle strikes [44], voltage droops [53], manufacturing defects [55], and/or aging effects [36]. This is particularly noticeable *at scale*, where even small error rates in hardware components can cause data corruptions that affect large-scale corporations such as Google [23] or Meta [13], prompting these corporations to deploy major resources to address the issue [56]. The problem of *silent data corruptions*, or SDCs, is further exacerbated in safety-critical domains (such as in autonomous vehicles, medical devices, or robotics), where even a few errors can lead to fatal and undesirable consequences.

At the same time, the rise of ML algorithms proliferating data centers and safety-critical domains means that hardware robustness to this particular domain is extremely important. Recent work has shown that as little as a single (non-adversarial) bit flip during a DNN's inference can cause wrong predictions by the model [34]. In the context of a self-driving car, this can potentially lead to consequential downstream decision making which can be fatal, such as accelerating instead of braking (e.g., by classifying a truck as a bird for instance) [34]. Thus, it is imperative to understand and mitigate the effect of *hardware* bit flips on an executing *software* application, where our focus here is on vision classification for its broad applicability and importance.

The current state-of-the-art technique to mitigate hardware errors is full modular hardware redundancy. This is the approach taken recently by Tesla in their Full Self-Driving (FSD) chip, where they employ a fully redundant co-processor with additional wiring, logic, and packaging to run two parallel inferences

37th Conference on Neural Information Processing Systems (NeurIPS 2023).

for comparison (and rerunning on a mismatch) [64]. While this approach is effective in identifying and mitigating errors during inference, the associated $2\times$ overhead is excessive and potentially unscalable for many domains which may need to operate under stricter hardware, energy, and cost budgets.

While memory errors can be protected by traditional error-correcting code (ECC) or additional parity bits at a fraction of the cost of full redundancy, errors that occur during *computation* are more difficult to address at low cost. Further, they are also exceptionally difficult to detect, since they are many times *silent* and do not cause an application to crash but still result in incorrect outcomes. Recent research at the intersection of ML and silent data corruption (SDC) detection has explored the use of low-cost dynamic range detectors during deployment [7], selective feature-map duplication [40], and inference re-execution [41] to detect and mitigate single-bit flips at run time in order to avoid full modular redundancy while targeting high error coverage. While these techniques have shown promise, they are more reactive in that they target *inference*, with the objective of hardening a pre-existing model to function in a faulty environment. Instead, in this work, we introduce what we believe is the first *training*-side technique, with the objective of developing out-of-the-box models that are more resilient against transient bit-flips in hardware.

In this paper, we present a novel *software-driven* solution to improve hardware reliability in neural networks. Our approach combines textual information from the Contrastive Language-Image Pre-training (CLIP) [49] model with visual information from a traditional classification neural network to strongly attenuate the effect of single-bit hardware errors in the computational components of a model. The proposed method is based on the observation that textual information can often provide useful context to interpret visual data, thereby enhancing the accuracy of error detection and correction. Our experiments show that the combination of textual and visual information can improve the reliability of a neural network's classification layer by up to $14\times$ compared to traditional error detection and correction techniques, with minimal changes to pre-existing training recipes and their corresponding training accuracy.

The primary contributions of this paper are:

1. **Contribution 1:** We propose a simple training methodology combining textual and visual information about an image to improve a model's robustness to hardware-based, transient computational errors which can occur during model deployment (§4).

2. **Contribution 2:** We rigorously evaluate our proposed methodology using both traditional accuracy-based metrics from the ML community, as well as reliability-based metrics from the hardware resiliency community§5. Our results provide a favorable tradeoff, where, on average, a $0.32\%$ validation accuracy loss on the ImageNet dataset translates to a hardware reliability improvement of up to $14\times$ (§6). Furthermore, we show that the $0.32\%$ average accuracy loss is statistically insignificant by analyzing the statistical distribution of predictions across both the original, unhardened model and our novel, robust model (§7).

3. **Contribution 3**: We provide a thorough discussion based on state-of-the-art visualization techniques and empirical data to explain why our method performs better, with ablation studies, intuitive explanations, and statistical validation (§7).

## 2   Scope and Limitations

This work is *not* about adversarial robustness, but rather focuses on hardware-based fault mitigation and analysis. Two high-level distinctions between these two are that (1) we do not assume a malicious adversary, but rather environmental effects which cause faults to occur in the hardware during the execution of a model [54, 33, 12, 58, 28, 6, 5], and (2) adversarial attacks typically corrupt the *input* of a model, while we focus on *computational* errors (i.e., neuron corruptions), which may occur in multiply-and-accumulate (MAC) operations during execution due to environmental- or manufacturing-based effects.

In the context of safety-critical systems and/or large-scale systems, these hardware errors are important to identify and mitigate to avoid data corruption at-scale, or fatally worse outcomes in real-time systems. While we believe the concept of resilience may be similar to adversarial robustness or Out-of-Domain (OOD) reliability (and in fact, our idea to use CLIP stems from this similarity), we focus our evaluation on improving hardware reliability. Exploring the correlation between *hardware reliability* and these other domain-specific reliability concepts is of particular interest for future work.

# 3 Related Work

**Unimodal Vision Models:** The AlexNet [31] model, introduced in 2012, was a Convolutional Neural Network (CNN) that gained popularity by winning the ImageNet [10] competition. It was followed by other models like VGG [57], which emphasized smaller filter sizes and deeper networks, and ResNet [20], which addressed the vanishing gradient problem with skip connections, enabling the training of very deep networks. More recent models such as EfficientNet [59], and MobileNet [24] have further improved efficiency and accuracy by utilizing compound scaling and lightweight architectures for mobile devices. However, CNNs have limitations such as limited receptive field and spatial inductive biases. To overcome these limitations, transformer-based approaches have emerged in computer vision. Inspired by the Transformer [61] architecture in natural language processing, the Vision Transformer (ViT) [14] model was proposed. It processes image patches as sequences and achieves competitive performance on various benchmarks. Other studies, like the Swin Transformer [38, 37] and MaxViT [60], have built upon the success of ViTs, focusing on improving accuracy and computational efficiency. Additionally, there are hybrid works that take inspiration from both Transformers and CNNs, such as FocalNets [65], which propose an efficient alternative to the self-attention operator, focal modulation, based on Convolutions. These models are typically trained using a cross-entropy objective. However, they have shown high susceptibility and unreliability to hardware errors [45, 51, 34, 35, 39], such as bit flips in the weights and activations. To ensure trustworthy deployment for real-world applications, it is crucial to establish strong resilience and reliability.

**Multi-Modal Vision-Language Models:** Advances in Natural Language Processing (NLP) has led to the development of vision-language models like CLIP [49], Align [26], and Florence [66]. These models consist of image and text encoders and are trained using a contrastive approach with extensive image-text pairs. The goal is to establish a shared feature space between visual and textual features, allowing models like CLIP [49] to gain a nuanced understanding of visual concepts. This approach benefits various downstream tasks such as "zero-shot" image classification, semantic segmentation [50, 17, 68], object detection [15], point cloud classification [67], and video recognition [46]. Additionally, CLIP has demonstrated impressive generalization capabilities on out-of-distribution tasks, including evaluations on datasets like ImageNet-A [22], ImageNet-R [21], ImageNet-Sketch [62] and ImageNetV2 [52]. However, training a CLIP model from scratch is prohibitively expensive. To address this, researchers have employed techniques like using enriched text prompts from large language models [48] such as GPT-3 [4] or employing prompting/finetuning methods [70, 69, 27, 46, 63] to enhance performance on out-of-distribution tasks.

The impact of such text-guided classification on hardware resilience and reliability is an unexplored topic in literature. The strong generalization capabilities demonstrated by text-guided classification models like CLIP suggest the potential for improved resilience to hardware errors. By leveraging the semantic supervision provided by text, these models can acquire a nuanced understanding of visual concepts, which may help them to better handle and adapt to errors or inconsistencies in hardware.

**Hardware Resilience and Reliability:** As NN-based image classification models begin to take off in vision-based tasks (such as for autonomous driving, or AV, systems), their robustness to hardware perturbations has become of paramount importance for practical deployment and government certification [25]. For example, Tesla's FSD uses the simplistic full duplication method to achieve high resilience, effectively allocating double the silicon to detect and correct errors [64]. However, due to the high associated costs of full modular duplication, an open call for cheaper yet equally accurate techniques has surfaced in recent years [45].

Rather than relying on hardware solutions for hardware faults, software-based solutions have risen to prominence due to their comparatively lower overhead in this fast-moving field. Proposed techniques leverage unique insights about the application domain (namely, neural networks) to systematically detect and recover from errors. Selective duplication of feature maps in CNNs [40], value attenuation in the form of range-detectors [7], and temporal re-execution of inferences [41] have all shown to be adept at identifying errors in a low-cost manner, with reasonable guarantees on error coverage.

However, all prior research in the field assumes a model is already trained and ready for deployment, and only then does the task of making it more resilient to hardware errors come into play (by using some of the aforementioned techniques above). In contrast to prior work, our focus in this paper is to provide a training routine that *generates* robust models directly using textual-visual information (§4). Effectively, our technique is a *training-based* method for designing robust image classification models, evaluated for its robustness to single-bit, transient hardware errors at run-time.

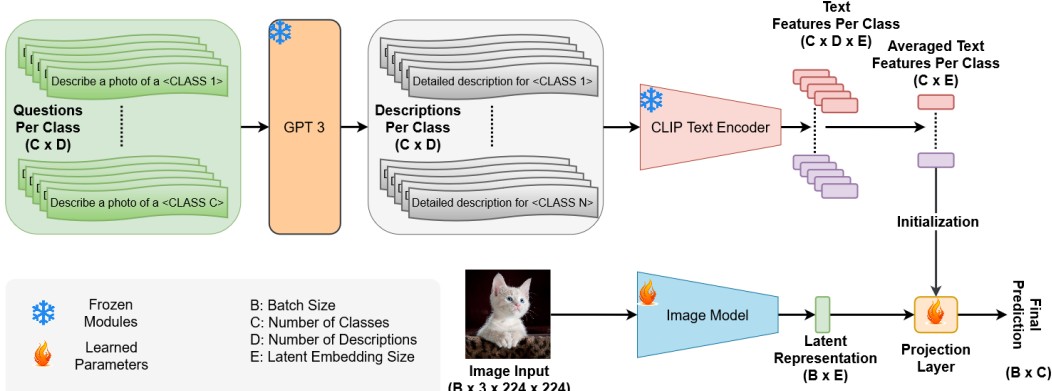

Figure 1: **The proposed Architecture:** $D$ questions for each class (total $C$) are hand-crafted. These are fed to a GPT-3 [4] model, to obtain $D$ detailed descriptions per class. A CLIP text encoder is used to produce text embeddings, which are averaged across descriptions. The text embeddings initialize a projection layer which is then trained alongside the randomly initialized backbone.

## 4 Our Approach

Multimodal pretrained models like CLIP [49], which are used for image classification, have demonstrated the ability to learn generalized representations. These models are trained on vast datasets of language-image pairs in a contrastive manner, resulting in impressive zero-shot capabilities and effective transfer learning to various downstream tasks.

Given this strong generalization, we ask the following question: *can the generalized representations of these Vision-Language models help improve hardware reliability?*. Our method augments the standard training of image classification models by utilizing textual context from the CLIP text encoder, allowing us to improve hardware resilience with minimal train and test time overhead.

We start by providing a brief overview of vision-language pre-training, specifically focusing on CLIP in §4.1. However, our methodology can be applied to other vision-language models that share similarities with CLIP, such as ALIGN and Florence. Following the overview, we provide a detailed explanation of our text-guided classification scheme in §4.2.

### 4.1 Overview of the CLIP Model

Conventional methods of image classification have traditionally used the common Cross-Entropy loss-based training for a closed-set classification problem [57, 20, 38, 37]. However, recently there has been a trend to employ text supervision for image classification rather than one-hot labels such as major works on contrastive language-image pretraining like CLIP [49]. The CLIP model is composed of two encoders that encode the visual content of images and their corresponding text descriptions, respectively. These encoded representations are then compared using a cosine similarity objective.

### 4.2 Proposed Text-Guided Classification

Consider an input image $I \in \mathbb{R}^{H \times W \times 3}$ of spatial size $H \times W$ with 3 channels Red, Green, and Blue (R, G, and B). A standard image classification model [57, 20, 37] maps the input image to the classification domain $\mathbb{R}^C$, where $C$ is the number of classes. However, it has been shown that such a model is unreliable and susceptible to bit errors [8, 41], especially in the classification layer [43].

To counter this problem we propose our text-guided image classifier which modifies the last layer of the image classification model to incorporate text features. Hence, this modification applies to any image classification model, regardless of the underlying architecture. Given an input image $I \in \mathbb{R}^{H \times W \times 3}$, we first map it to a latent dimension $\mathbb{R}^E$ where $E$ is the embedding length of the CLIP Text Encoder. We then apply a classification projection $\mathbf{P}_{class}$ which maps the latent dimension $\mathbb{R}^E$ to $\mathbb{R}^C$. We initialize the projection layer $\mathbf{P}_{class}$ using features obtained from the CLIP text encoder for each class.

A naive way to obtain the text features would be to simply pass each class name through the text encoder. However, this is less robust to distribution shifts [70], and we argue that it would therefore be less reliable. Instead, we follow [48] and augment the class labels using a large question-answering language model like GPT-3 [4]. We ask GPT-3 a total of $D$ questions for each class in the total number of classes $C$, making a total of $C \times D$ questions. For each question, GPT-3 outputs detailed text descriptions, forming $D$ number of descriptions per class, which we then pass through the CLIP text encoder to produce embeddings of shape $C \times D \times E$. We then average over the descriptions per class, to form the final embedding tensor of shape $C \times E$, where each class $c \in 1,...,C$ has an embedding vector in $\mathbb{R}^E$. This tensor is then used to initialize the projections layer $\mathbf{P}_{class}$. Figure 1 summarizes our proposed approach.

## 5  Evaluation Methodology

We evaluate our approach on two fronts: first, the impact of our technique on the classification accuracy of the model; and second, the hardware reliability impact of our new technique.

### 5.1  Evaluation Infrastructure

For each backbone, we train a baseline model (the default architecture), and our modified classification model on the ImageNet training set. For both methods, we follow the standard PyTorch [47] training recipe. We then report accuracy and resilience on the ImageNet validation set and compare across multiple architecture families. We train our models on $4 \times$A100 GPUs and run the training routine for both the original and our modified models to the same number of epochs and with the same hyperparameters as described in the PyTorch Github repository [1] for a fair comparison. Our results are presented in §6, in Table 1. For the ablation study presented in Table 3, we train each model on $8 \times$V100 GPUs.

### 5.2  Hardware Reliability Evaluation Methodology

To evaluate the reliability of the proposed model compared to the original model, we use the GoldenEye [43] testbed for error analysis. We describe how this testbed works in more detail in this section. Due to the exponentially large number of potential hardware error sites (e.g., a single bit flipping in a random register during any dynamic cycle for any image during inference at deployment time), it is impractical to explore all possible error locations and values for a given error model to perform an exhaustive evaluation of a DNN. Instead, *error injection* mechanisms are used to statistically evaluate the likelihood of errors propagating and corrupting an application's output [34, 8, 39, 40].

In this work, we use a transient single-bit flip error model for evaluation, a commonly used abstraction for modeling hardware faults [34, 18, 35, 2, 19]. In particular, we focus on errors that occur in *activation values* (i.e., neurons), during inference. We assume that memory is protected via ECC or parity (which is common in commercial and safety-critical systems [42]), allowing us to focus on computational errors in hardware (i.e., MAC operations).

An error injection experiment involves flipping a single bit in the entire network (i.e., from 0→1 or 1→0), and then comparing the final classification of the network with respect to the original, baseline correct output. We use PyTorchFI [39] to perform the random bit flip, and we perform 4096 unique error injection experiments per layer, totaling more than 4.3 million experiments across all our models and corresponding to a 99% confidence level with less than 0.23% confidence interval [32].

To measure reliability, we calculate the rate of all errors which led to an image misclassification, as a function of all the injections performed. Recent work [41] has proposed a more accurate metric called $\Delta$Loss, which captures the same information as mismatches but converges asymptotically faster. Conceptually, the $\Delta$Loss metric calculates the difference in cross entropy (CE) between a fault-free inference and an erroneous inference, rather than purely looking at a binary mismatch. Consequently, it provides more granular information per error injection experiment. We use the $\Delta$Loss metric for comparing the reliability of each layer in the network for the original, baseline training routine and our proposed, textual-augmented technique. To gather overall network-level reliability improvement, we average the $\Delta$Loss information gathered from each layer, producing a singular value to compare the baseline model and our proposed text-guided model. We note that it is a simple mapping to go

---

[1]https://github.com/pytorch/vision/tree/main/references/classification

back to the mismatch-based metric and ground this in a hardware-centric FIT-rate, while this work leverages the $\Delta$Loss metric for it's drastically faster speed and accuracy [41].

Prior work has shown that the activation values of the final layer of a network are typically the most vulnerable to single-bit perturbation, as this layer directly impacts the final classification [43]. For this reason, we target our technique and analysis on the last layer, in an effort to mitigate errors at this stage. To ensure a fair comparison, we compare the last layer of the baseline model with the weighted average of the last two layers of our proposed model. This is because our proposed technique technically splits the original last layer into two fully connected layers: the latent representation (B×E) and a projection layer (E×C). We combine these last two layers into a single layer (B×C) for efficiency and a fair head-to-head evaluation during inference (we keep them separate during training for initializing the projection layer).

We further show the necessity of the rich GPT-3 initialization of our method through an ablative study in Table 3. Finally, we provide a qualitative analysis using Ablation-CAM [11] as well as quantitative analysis for per-image inferences in §7 to further validate the benefits and trade-offs of our textual-visual-based approach for improved hardware reliability. We use the concept of the *Top2Diff* from the hardware reliability literature [41] to build intuitive arguments on the reliability of our model. The Top2Diff metric is simply the difference in classification accuracy between the top inferred class and the second-highest class. A large Top2Diff directly translates to better reliability, as the catalyst required by a single bit flip to overcome and flip the classification from the correct class to a different class is larger.

## 6 Results

Our main results across various backbones are summarized in Table 1. For each model backbone, we trained a baseline version using the training recipe provided by PyTorch, followed by our own method trained using the text-guided initialization via GPT-3, again with the same recipe. We used the same set of hyperparameters for both models (detailed hyperparameters are reported in Appendix A). We report the accuracy of the baseline model, the accuracy of our proposed approach, the difference in the number of parameters of the two versions, the difference in FLOPs, and the improvement in reliability on the last layer and across the entire backbone. We observe multiple takeaways in our results, which are described below.

Table 1: The tables present results across various backbones, reporting top-1 accuracy on the ImageNet [10] validation set for both the baseline and our method. Additionally, we report the change in total parameters and FLOPs (a negative sign indicates a decrease), the improvement in last-layer and overall model reliability, and a percentage increase in Top2Diff.

| Backbone | Acc. Baseline | Acc. Ours | Additional Params (w.r.t baseline) | Additional FLOPs (w.r.t baseline) | Improvement in Reliability (Last Layer) | Improvement in Reliability (Overall) | Improvement in Top2Diff |
|---|---|---|---|---|---|---|---|
| Alexnet [31] | 56.43% | 57.28% | $-1.49M$ | $-1.50M$ | 7.92× | 4.67× | 2.83% |
| VGG-16-BN [57] | 73.45% | 72.96% | $-1.49M$ | $-1.50M$ | 14.43× | 9.64× | 1.62% |
| VGG-19-BN [57] | 74.40% | 74.01% | $-1.49M$ | $-1.50M$ | 13.29× | 8.67× | 1.13% |
| ResNet-18 [20] | 69.60% | 69.68% | $0.26M$ | $0.26M$ | 2.87× | 1.91× | 3.07% |
| ResNet-34 [20] | 73.25% | 72.62% | $0.26M$ | $0.26M$ | 3.89× | 2.53× | 2.08% |
| ResNet-50 [20] | 75.64% | 74.84% | $-0.49M$ | $-0.49M$ | 4.48× | 2.96× | 3.35% |
| ResNet-101 [20] | 77.25% | 75.52% | $-0.49M$ | $-0.49M$ | 4.33× | 2.77× | 3.13% |
| ResNet-152 [20] | 77.98% | 76.18% | $-0.49M$ | $-0.50M$ | 4.47× | 2.85× | 3.09% |
| MobileNet-V2 [24] | 71.87% | 71.83% | $-0.11M$ | $-0.09M$ | 3.92× | 2.43× | 5.36% |
| MaxViT-T [60] | 82.98% | 83.08% | $0.26M$ | $0.28M$ | 3.38× | 2.63× | 2.62% |
| Swin-V2-T [37] | 80.97% | 80.02% | $0.13M$ | $0.15M$ | 1.65× | 1.07× | 2.85% |
| Swin-V2-S [37] | 82.71% | 82.86% | $0.13M$ | $0.15M$ | 3.51× | 2.60× | 3.04% |
| FocalNet-T [65] | 80.23% | 80.77% | $0.13M$ | $0.14M$ | 3.87× | 2.61× | 2.61% |
| FocalNet-S [65] | 82.01% | 82.52% | $0.13M$ | $0.14M$ | 4.73× | 3.50× | 3.10% |

**Accuracy impact:** Our proposed model has a small accuracy reduction on the backbone compared to the baseline, ranging from $-1.77\%$ to $+0.52\%$, for an average decrease of $.3\%$. Despite the reduction, we find that our proposed model is in fact *more* confident in its accurate predictions based on the difference between the top two classes (the *Top2Diff*). For ResNet50, the Top2Diff for the baseline model is 70.32%, while the Top2Diff of our model is 73.67%, a $+3.35\%$ improvement. A similar phenomenon is observed across all models, where the average Top2Diff increases by $2.50\%$. Further, we perform an additional study in §7.4, where we empirically show that this accuracy impact is indeed minimal, especially compared to the upside observed in reliability improvement.

Table 2: The tables present results across various datasets (CIFAR10 [29], CIFAR100 [30], FOOD101 [3], and STL10 [9]) for two backbones (ResNet-50 [20] and FocalNet-T [65]), reporting top-1 accuracy on the respective validation set for both the baseline and our method. Additionally, we report the improvement in last-layer and overall model reliability, and a percentage increase in Top2Diff.

| Dataset | Backbone | Acc. Baseline | Acc. Ours | Improvement in Reliability (Last Layer) | Improvement in Reliability (Overall) | Improvement in Top2Diff |
|---|---|---|---|---|---|---|
| CIFAR10 [29] | ResNet-50 [20] | 95.07% | 95.29% | 2.04x | 1.71x | 6.70% |
| CIFAR10 [29] | FocalNet-T [65] | 94.76% | 94.94% | 2.47x | 1.30x | 3.58% |
| CIFAR100 [30] | ResNet-50 [20] | 78.23% | 78.53% | 2.19x | 1.65x | 3.69% |
| CIFAR100 [30] | FocalNet-T [65] | 77.06% | 79.21% | 3.21x | 1.58x | 2.90% |
| FOOD101 [3] | ResNet-50 [20] | 83.13% | 83.97% | 2.66x | 2.15x | 2.78% |
| FOOD101 [3] | FocalNet-T [65] | 85.64% | 85.91% | 3.28x | 2.85x | 1.70% |
| STL10 [9] | ResNet-50 [20] | 47.73% | 52.68% | 2.10x | 1.91x | 2.45% |
| STL10 [9] | FocalNet-T [65] | 62.74% | 63.78% | 2.23x | 1.72x | 1.96% |

**Model Size and Runtime Impact:** Our proposed method marginally increases the total number of parameters, on average, by $0.18M$ compared to the baseline. This reduction is model-dependent, as it depends on the second-to-last layer feeding into the latent representation ($B \times E$) before moving onto the projection layer ($E \times C$) for the final prediction. A few models (such as deeper ResNet's and VGG) actually observe a slight *decrease* in total model parameters, which is topology dependent. This parameter difference translates to a small increase/decrease of FLOPs during inference, respectively, as show in column 5. Overall, our proposed technique produces models with similar size and runtime to the baseline on average.

**Evaluation on Additional Datasets:** We evaluate our method on additional datasets (CIFAR10 [29], CIFAR100 [30], Food101 [3], and STL10 [9]) for two networks: ResNet-50 [20] and FocalNet-T [65].. Our results, shown below in Table 2, validate that our technique is general and can work across an array of model types and datasets. Furthermore, we did not have to modify any hyperparameters in the process, suggesting the ease of our technique as well as the increased benefit from a reliability point of view. Additionally, adding these new datasets further support our claims that our technique has negligible impact on model training accuracy, whilst still providing us with a large upside in resilience.

**Reliability Evaluation:** Most importantly, our proposed technique significantly improves the hardware reliability of the model, as this was the intention behind the method. The most significant change occurs on the last layer, where the average reduction is model-family specific. In other words, the ResNet family observes an average $4.01\times$ hardware reliability improvement and the VGG family observes a $13.68\times$ improvement on the final layer, using the $\Delta$Loss metric as explained in §5. This difference is related to the baseline backbone, where in general the ResNet family baseline can be considered more reliable than the VGG family to hardware errors [41], resulting in a larger improvement with our proposed technique for the less robust model family (VGG). Overall, we observe improvements across the board for all models studied, signifying the benefits of our technique. Similarly, looking at a model's end-to-end hardware reliability, we find the average to be $9.16\times$ better for the VGG family, and $2.61\times$ for the ResNet family. While this value is strongly influenced by the last layer, we observe that most layers in the network do get a modest hardware resilience improvement, captured by the averages listed in the table and additional results in Appendix C.

## 7    Discussion and Analysis

We perform a series of additional studies to validate and better understand the insights of our proposed method. First, we describe an ablation study on the initialization of the projection layer in §7.1. Second, we provide a qualitative explanation of the impact of errors on the baseline versus our proposed method using the state-of-the-art Ablation-CAM visualization in §7.2. Third, we analyze the impact of the baseline training versus our method's training on the activation values produced by each model, and use this to provide an intuitive explanation for the improved hardware reliability in §7.3. Finally, in §7.4, we further discuss the tradeoff between the small accuracy drop on the validation set compared with the large improvement in hardware reliability by studying the output classification accuracy of images.

## 7.1 Ablation Study

Our ablation study (Table 3) measures the improvement in hardware reliability for different projection initialization techniques on ResNet50. We compare 1) a random initialization, 2) a CLIP-based initialization ("a photo of a [CLASS]" prompt), and 3) our CLIP+GPT-3 initialization.

Table 3: Ablation for type of initialization on the projection layer. CLIP refers to a simple hand-crafted prompt "a photo of a [CLASS]" while CLIP+GPT refers to the proposed method in §4.

| Backbone | Projection Initialization | Improvement in Reliability (Last Layer) | Improvement in Reliability (Across Backbone) |
|---|---|---|---|
| ResNet-50 | random | $1.74\times$ | $1.09\times$ |
| ResNet-50 | CLIP | $5.06\times$ | $3.28\times$ |
| ResNet-50 | CLIP+GPT-3 | $6.09\times$ | $3.93\times$ |

In general, we find that any text-based projection initialization helps improve reliability, as observed via the "random" experiment which gives a $74\%$ last layer improvement, and an overall $9\%$ improvement across the network compared to the baseline. However, a more intelligent projection initialization via CLIP improves the hardware reliability up to $3.28\times$ across the network ($5.06\times$ for the final layer) and good prompting via GPT-3 to "describe a [CLASS]" further improves it to $3.93\times$ ($6.09\times$ for the final layer). To summarize, our ablation study validates that our good hardware reliability improvements indeed come from our proposed initialization.

## 7.2 Ablation-CAM Visualization

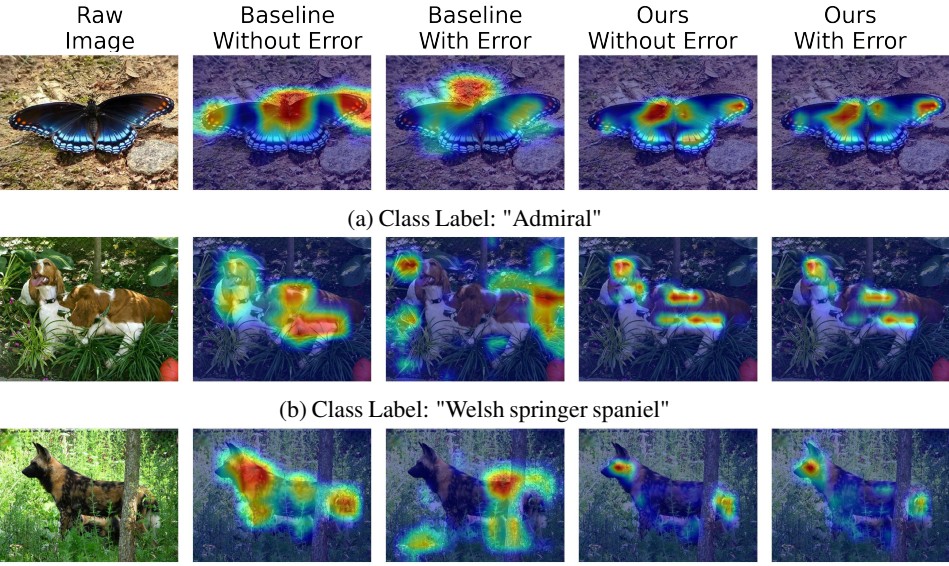

(a) Class Label: "Admiral"

(b) Class Label: "Welsh springer spaniel"

(c) Class Label: "African hunting dog"

Figure 2: Comparative Visualization of Baseline and Our version of ResNet-50 Before and After Single Bit Flip Error Injection. Each subfigure presents the original image, alongside the Class Activation Mapping (CAM) visualizations for both the baseline and our model before and after error injection. Prior to error injection, both models concentrate on key features of the images for classification. Post error injection, the baseline model's focus diverges, for instance, from the African hunting dog to the surrounding foliage (Figure 2c), whereas our model maintains its original focus, demonstrating its robustness against the induced error.

Ablation-CAM is a visualization technique developed for DNNs that employs a gradient-free approach [11]. A departure from conventional gradient-based methods, Ablation-CAM systematically removes parts of the network, allowing a deeper understanding of the individual feature map units

contributing to the model's decisions. This ablation process generates a coarse localization map, highlighting regions in the network's input image that are critical for predictions.

In our study, we chose Ablation-CAM to visualize the decision-making process of our models (original versus our proposed technique). Ablation-CAM's gradient-free nature offers a more robust way of comprehending the focus areas within deep learning models, addressing the limitations of gradient-based methods, such as susceptibility to noise in the gradient and the inability to capture the entire network's collective decision-making process [1]. Furthermore, Ablation-CAM's ability to evaluate model trustworthiness [11] was critical in understanding the robustness of our models to error injection. By observing how the models' focus shifts in response to error injection, we could make judgments about their resilience and reliability. This unique combination of features made Ablation-CAM an ideal tool for our study.

Figure 2 depicts our results for three images on the ResNet50 backbone (Additional visualizations are presented in Appendix B). We inject 2000 random errors in weights across the network (for visualization purposes) and project the impact on the input image, to see how the model responds to the same exact perturbations. Our results highlight the fact that despite the many errors, our proposed technique maintains focus on the important features which correspond to better hardware-level reliability as discussed in §6.

### 7.3 Value Ranges

Another angle we study in this work is the impact of our proposed technique on the value of ranges for activation values (i.e., neurons) of a model. Prior work has shown that smaller values during computations typically are more robust to single-bit perturbations, as their impact does not propagate as far (except for errors in the exponent fields of a floating point value, which range detectors or quantization can help mitigate). In fact, Google previously proposed a ReLU6 [24] activation function to clip values above the arbitrary value of 6, later used as a range detector for reliability purposes [16]. Similarly, an organic reduction in values is beneficial from a hardware reliability perspective, which we target with our approach.

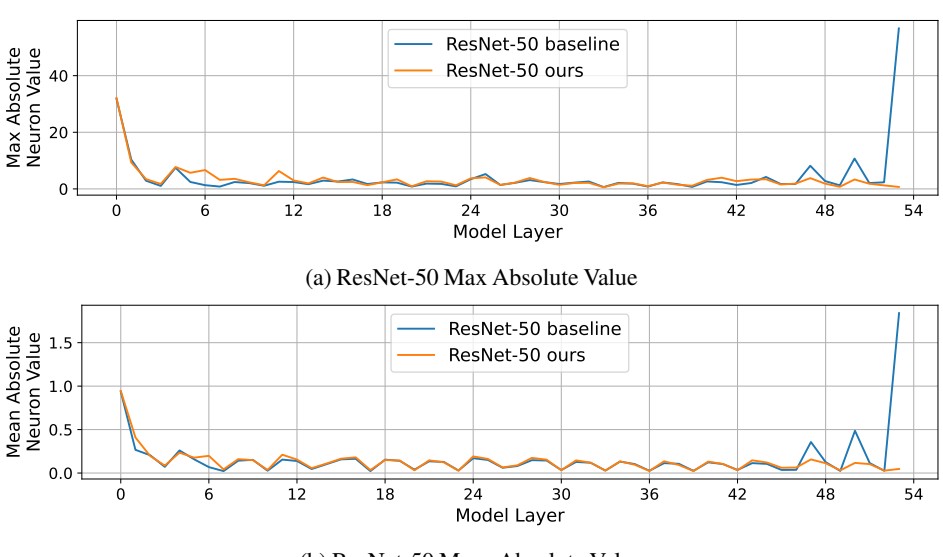

(a) ResNet-50 Max Absolute Value

(b) ResNet-50 Mean Absolute Value

Figure 3: **Observed Neuron values for ResNet50.** The Y-axis shows the max absolute value (Figure 3a) and mean absolute value (Figure 3b) observed by profiling the ImageNet dataset on the baseline and our model on a per-layer basis. It can be seen that both max and mean are viable choices for profiling network neuron value ranges, and result in similar trends.

Figure 3 shows the absolute maximum and absolute mean observed neuron values per layer for ResNet50 across the ImageNet dataset. We find that our proposed technique strongly attenuates the values in the last layer for both measurems. This result helps explain why our technique is more reliable, and why it is particularly beneficial for the final layer. The fault-free values are smaller to begin with, which in turn enable smaller range detectors [7] and also are less likely to change into a negative impactful error on the classification result. We observe a similar trend across all network studies in our experiments. We provide additional results for different networks in Appendix C.

The number representation in hardware also plays a large role in the reliability of a model, which our proposed technique directly influences. To better understand this effect, we direct the reader to the hardware implementation of numbers, which typically use the IEEE-754 floating point format, consisting of a sign bit, exponent bits, and mantissa bits. Intuitively, bit flips in the exponent bit are the most egregious, and having range detectors in place helps detect these types of errors. More subtle, however, is that depending on the original exponent value, certain bit flips can transform a number to become much larger or much smaller. In this case, a bit flip in a small number (which we identify as smaller than 2) has a very high probability of changing to *another* small value, regardless of which bit is flipped. In particular, so long as the sign bit or the most significant bit of the exponent (bit 31 and 30) are not the ones flipped, then the IEEE-754 format *guarantees* that the new, erroneous number stays under the value 2. As such, the small magnitude change has little impact on the end-to-end inference of a classification, and masks such errors. This is why it is crucial and advantageous to have smaller neuron values in a neural network, which various techniques such as batch normalization and our newly proposed technique help organically enforce (unlike an artificial enforcer such as ReLU6). Thus, our new training routine helps accomplish this through the use of the projection layer at the tail end of the neural network.

### 7.4    Understanding the Accuracy Degradation in the Context of Hardware Reliability

To better understand the $\sim 0.3\%$ accuracy loss of our technique, we wanted to see how the baseline and proposed models matched up if we excluded low-confidence images (i.e., images "at the border" during classification). In practice, low-confidence images would not be relied upon for safety-critical decision-making - hence we wanted to measure the "true" accuracy of the models.

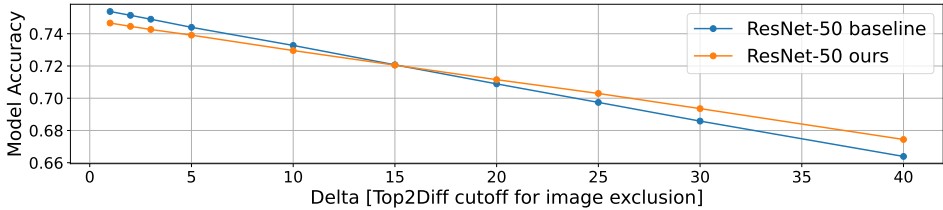

Figure 4: **Model accuracy as a function of Top2Diff deltas for ResNet50**. This figure shows that as we exclude images with low Top2Diff for classification accuracy measurements, our proposed model recoups accuracy faster than the original baseline model, indicating that many correctly classified images by the baseline model are borderline correct, to begin with.

We perform a sweep of different Top2Diff values (where "Delta" goes from 1% to 40%) and exclude the "correct" images that have a Top2Diff value below each sweep point. We measure the new network accuracy in light of these delta values and found that many images that were classified correctly by the original model "fell off" as we increased the delta. On the other hand, our proposed model did not lose its classification accuracy as fast; at a delta of Top2Diff=15, the inflection point occurs where our method has the same accuracy (i.e., $0\%$ accuracy degradation between models) as the original model, and improves beyond this point. That said, a $0.3\%$ accuracy loss itself is reasonable in and of itself for the large hardware reliability gains we observe, yet this discussion point presents a trade-off opportunity (as a function of Top2Diff) that can enable a model designer to tune their model for their desired accuracy and hardware reliability targets. To further validate this claim, we find that for different datasets (Table 2), our technique marginally improves accuracy across the board.

## 8    Conclusion

In conclusion, our paper presents a software-driven solution to enhance hardware reliability in neural networks. By combining textual and visual information, we mitigate the impact of transient bit-flips during computation. Our approach improves neural network reliability of the most critical layer by up to $14\times$ compared to the baseline, with minimal changes to training. We contribute a simple training methodology, rigorous evaluation using accuracy and reliability metrics, and a comprehensive discussion supported by visualization techniques. Our work highlights the significance of addressing hardware errors during training, offering a promising direction for developing robust models against transient bit-flips.

## 9 Acknowledgments

This work is supported in part by the National Science Foundation (NSF) grant CCF-1704834. We also thank the Fatima Fellowship for their GPU grant and Hugging Face for organizing and sponsoring the Fatima Research Fellowship program.

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
