# Supplementary Material
## Hardware Resilience Properties of Text-Guided Image Classifiers

This section contains supplementary material that provides additional details for the main paper and further experimental analysis. We include this content in the following order:

- Detailed Hyperparameters (Appendix A)
- Additional Visualizations (Appendix B)
- Additional Analysis (Appendix C)

## A    Detailed Hyperparameters

In this section, we provide detailed hyperparameters (Table 4) used to train each of the architectures on which results are reported in the main paper. Note that if the batchsize is reduced, the learning rate should be linearly scaled accordingly.

Note that for error injection experiments, we perform single-bit flips only in the convolutional and linear layers of the neural network, in line with other work in this field. The primary motivation is that these two layer types are the most computationally intensive, consuming $90\% - 95\%$ of a DNN's computations. Thus, these are the most likely locations for a hardware error to occur, and we focus our efforts on analyzing and evaluating the vulnerability in such layers.

Table 4: Training hyperparameters for different backbones presented in the main paper and supplementary material. "-" indicates that the particular method was not used at all.

| | Alexnet, VGG | ResNet | MobileNet-V2 | MaxViT | Swin-V2 | FocalNet |
|---|---|---|---|---|---|---|
| *Optimization* | | | | | | |
| Number of GPUs | 4×A100 | 4×A100 | 4×A100 | 4×A100 | 4×A100 | 4×A100 |
| Batch size (per GPU) | 180 | 180 | 180 | 1024 | 256 | 256 |
| Optimizer | SGD | SGD | SGD | AdamW | Adamw | Adamw |
| Epochs | 90 | 90 | 300 | 400 | 300 | 300 |
| Weight Decay | 0.0001 | 0.0001 | 0.0001 | 0.05 | 0.05 | 0.05 |
| Base learning rate | 0.01 | 0.1 | 0.045 | 0.003 | 0.001 | 0.001 |
| Minimum Learning Rate | 0.0 | 0.0 | 0.0 | 0.00001 | 0.00001 | 0.00001 |
| Learning rate schedule | StepLR | StepLR | StepLR | CosineAnnealingLR | CosineAnnealingLR | CosineAnnealingLR |
| Linear warmup epochs | 0 | 0 | 0 | 32 | 20 | 20 |
| Warmup method | - | - | - | Linear | Linear | Linear |
| StepLR Gamma | 0.1 | 0.1 | 0.98 | - | - | - |
| StepLR Step Size | 30 | 30 | 1 | - | - | - |
| *Data augmentation* | | | | | | |
| Random erasing probability | 0 | 0 | 0 | 0.25 | 0.25 | 0.25 |
| Train/Validation Crop Size | 224 | 224 | 224 | 224 | 256 | 224 |
| Label smoothing | 0 | 0 | 0 | 0.1 | 0.1 | 0.1 |
| Mixup ($\alpha = 0.8$) probability | 0 | 0 | 0 | 1.0 | 1.0 | 1.0 |
| Cutmix ($\alpha = 1.0$) probability | 0 | 0 | 0 | 1.0 | 1.0 | 1.0 |

# B   Additional Visualizations

In this section, we provide visualizations of additional backbones. Additional visualizations are provided for VGG-16-BN/VGG-19-BN (Figure 5), ResNet-18 (Figure 6), ResNet-34 (Figure 7) and MobileNet-V2 (Figure 8).

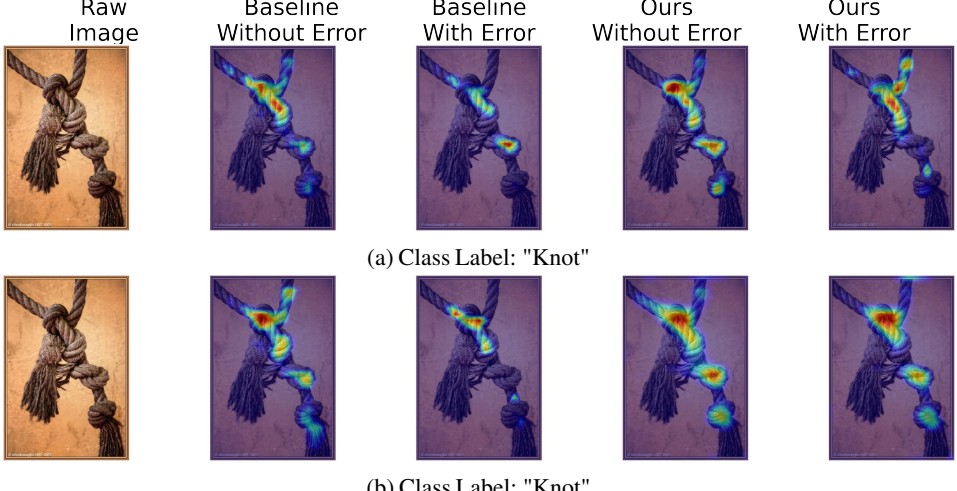

Figure 5: Comparative Ablation-Cam Visualization of Baseline and Our Models Before and After Error Injection on VGG-16-BN (Figure 5a) and VGG-19-BN (Figure 5b).

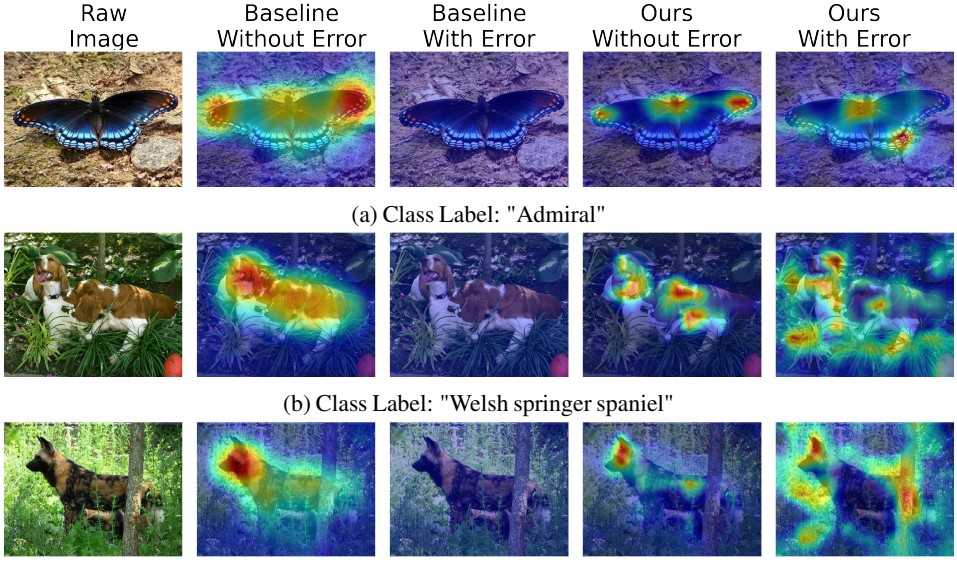

Figure 6: Comparative Ablation-Cam Visualization of Baseline and Our Models Before and After Error Injection on ResNet-18.

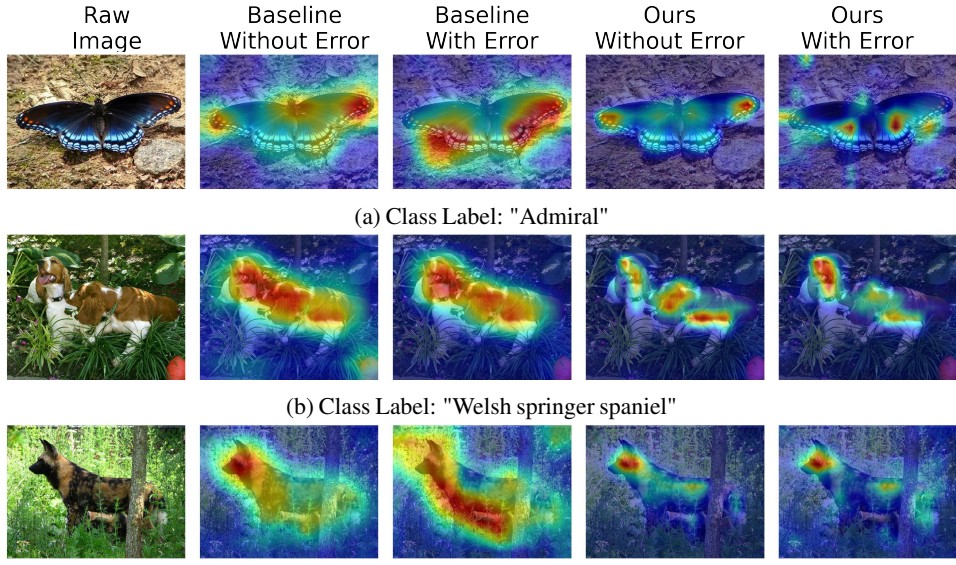

(a) Class Label: "Admiral"

(b) Class Label: "Welsh springer spaniel"

(c) Class Label: "African hunting dog"

Figure 7: Comparative Ablation-Cam Visualization of Baseline and Our Models Before and After Error Injection on ResNet-34.

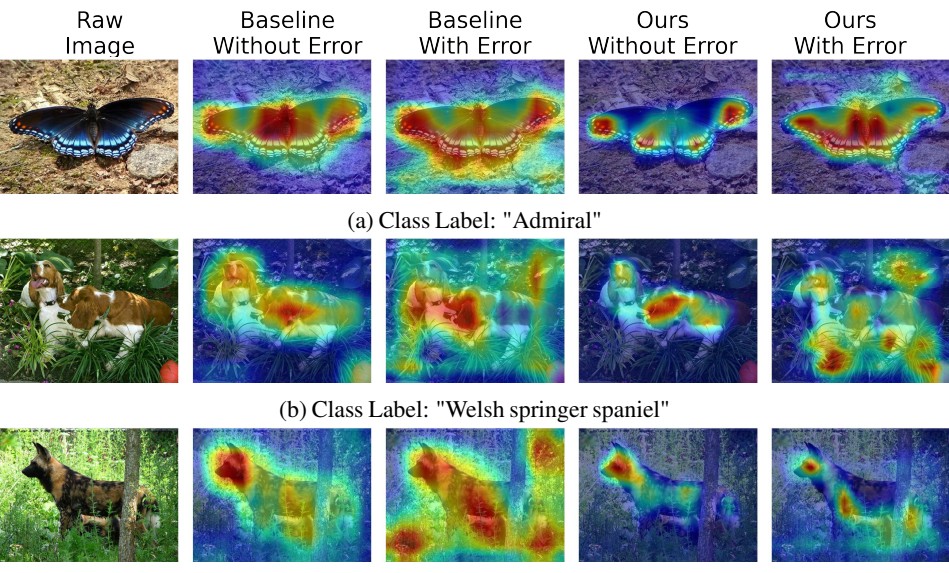

(a) Class Label: "Admiral"

(b) Class Label: "Welsh springer spaniel"

(c) Class Label: "African hunting dog"

Figure 8: Comparative Ablation-Cam Visualization of Baseline and Our Models Before and After Error Injection on MobileNet-V2.

## C  Additional Analysis

We provide additional data and results to extend our analysis from §7, providing information on other networks studied. The main takeaways and conclusions hold, and thus these additional plots and figures help reinforce our findings and comparison between our proposed technique and the baseline.

Figure 9 and Figure 10 extend from Figure 3 for more networks. The Y-axis shows the absolute value of the max neuron value observed per layer on the X-axis. As highlighted in §7, our proposed method helps organically attenuate the values observed at each layer, which translates to better hardware reliability.

Next, Figure 11 and Figure 12 are extensions for Figure 4, showcasing the impact of our proposed technique on the end-to-end network accuracy. Our results show that if we exclude low-confidence images from both the baseline model and our proposed model, our model holds onto classification accuracy more robustly. This is even pronounced for the Swin transformer model, where despite a marginal improvement in hardware reliability, its classification accuracy is better and more confident compared to the baseline model (see Figure 11e).

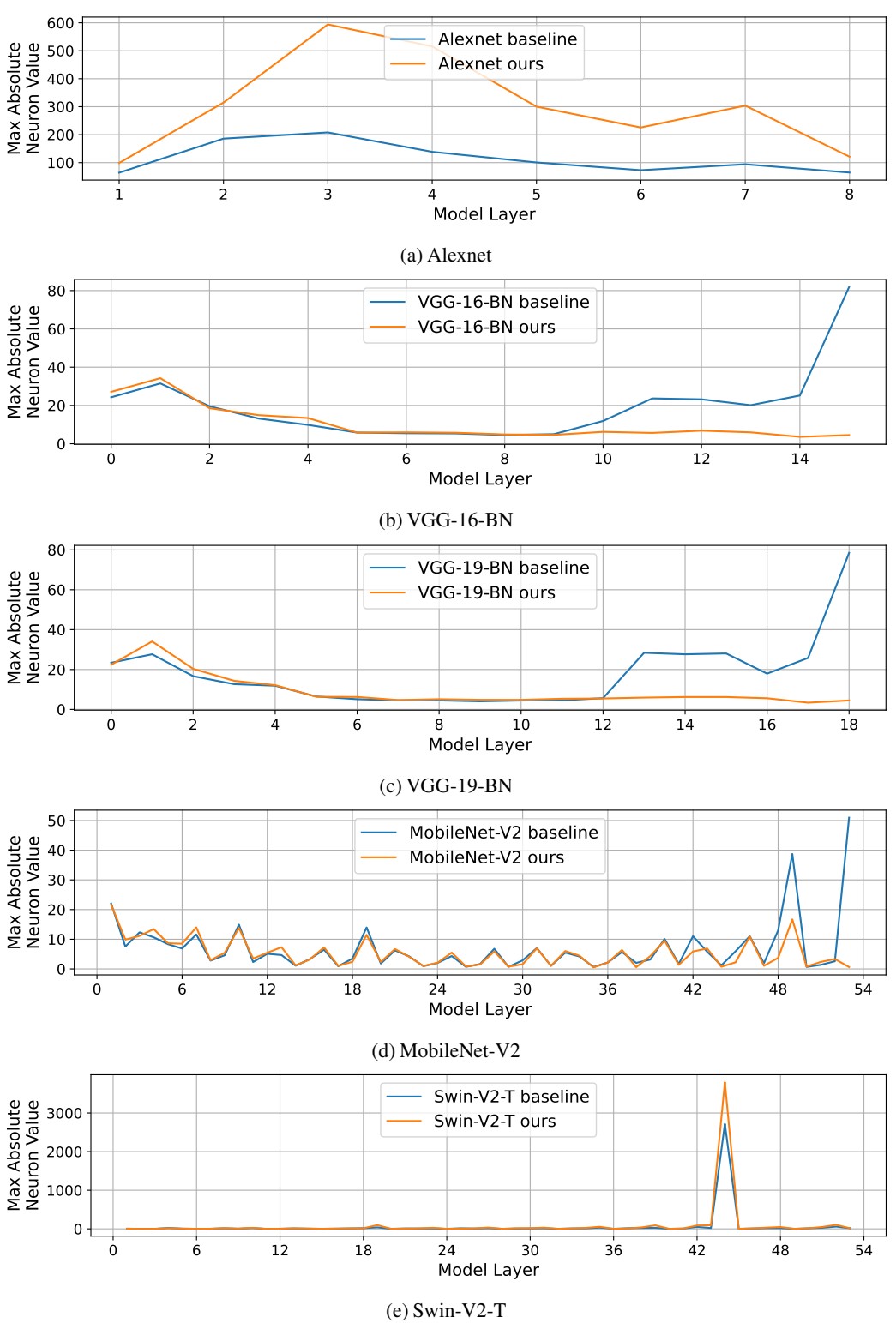

Figure 9: **Observed Neuron values.** This is an extension of Figure 3 for additional non-ResNet networks. As shown, our proposed method helps attenuate activation values across layers, particularly the last, critical layer. This in turn results in improved hardware reliability to single-bit errors.

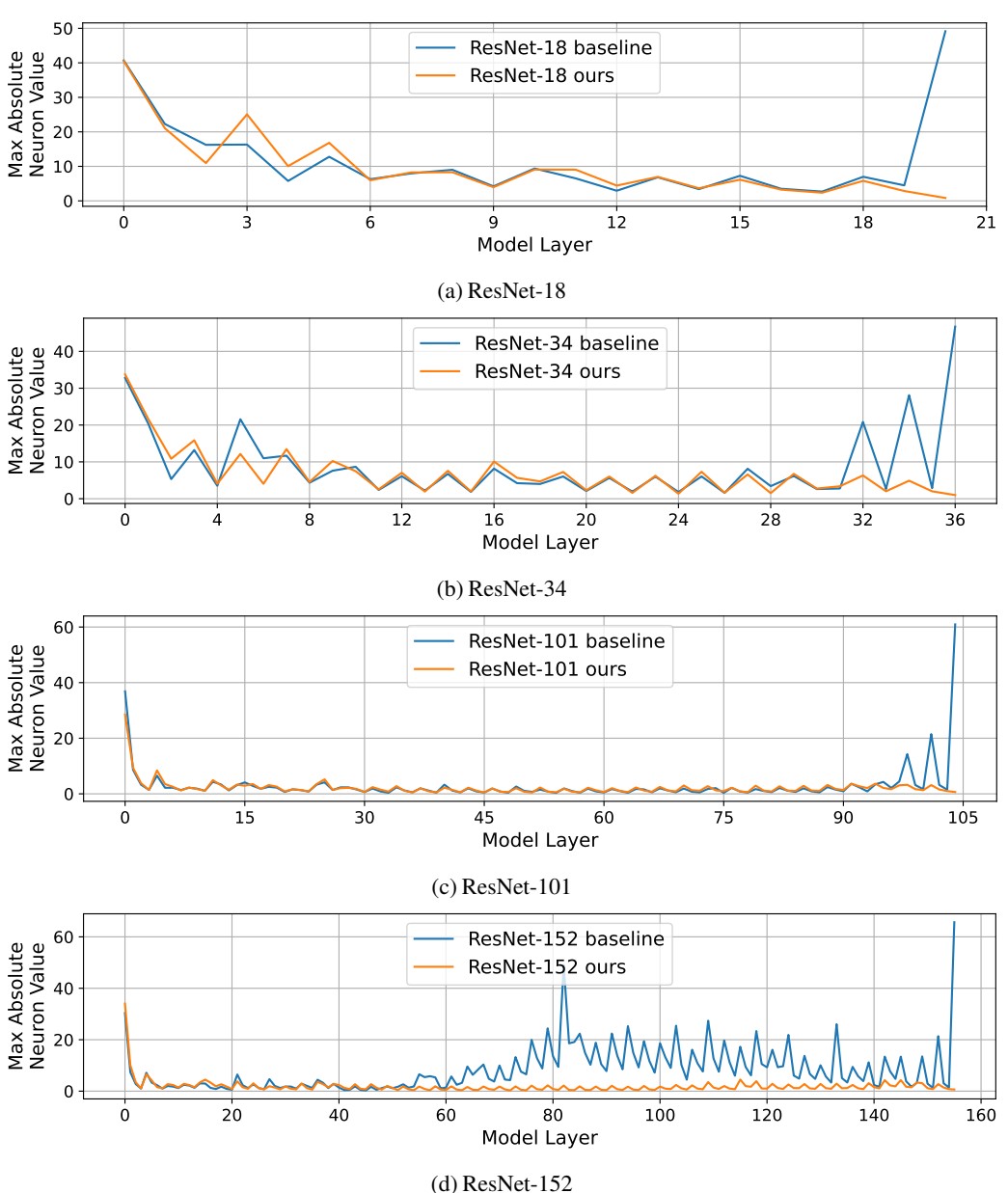

(a) ResNet-18

(b) ResNet-34

(c) ResNet-101

(d) ResNet-152

Figure 10: **Observed Neuron values.** This is an extension of Figure 3 for additional ResNet networks. As shown, our proposed method helps attenuate activation values across layers, particularly the last, critical layer. This in turn results in improved hardware reliability to single-bit errors.

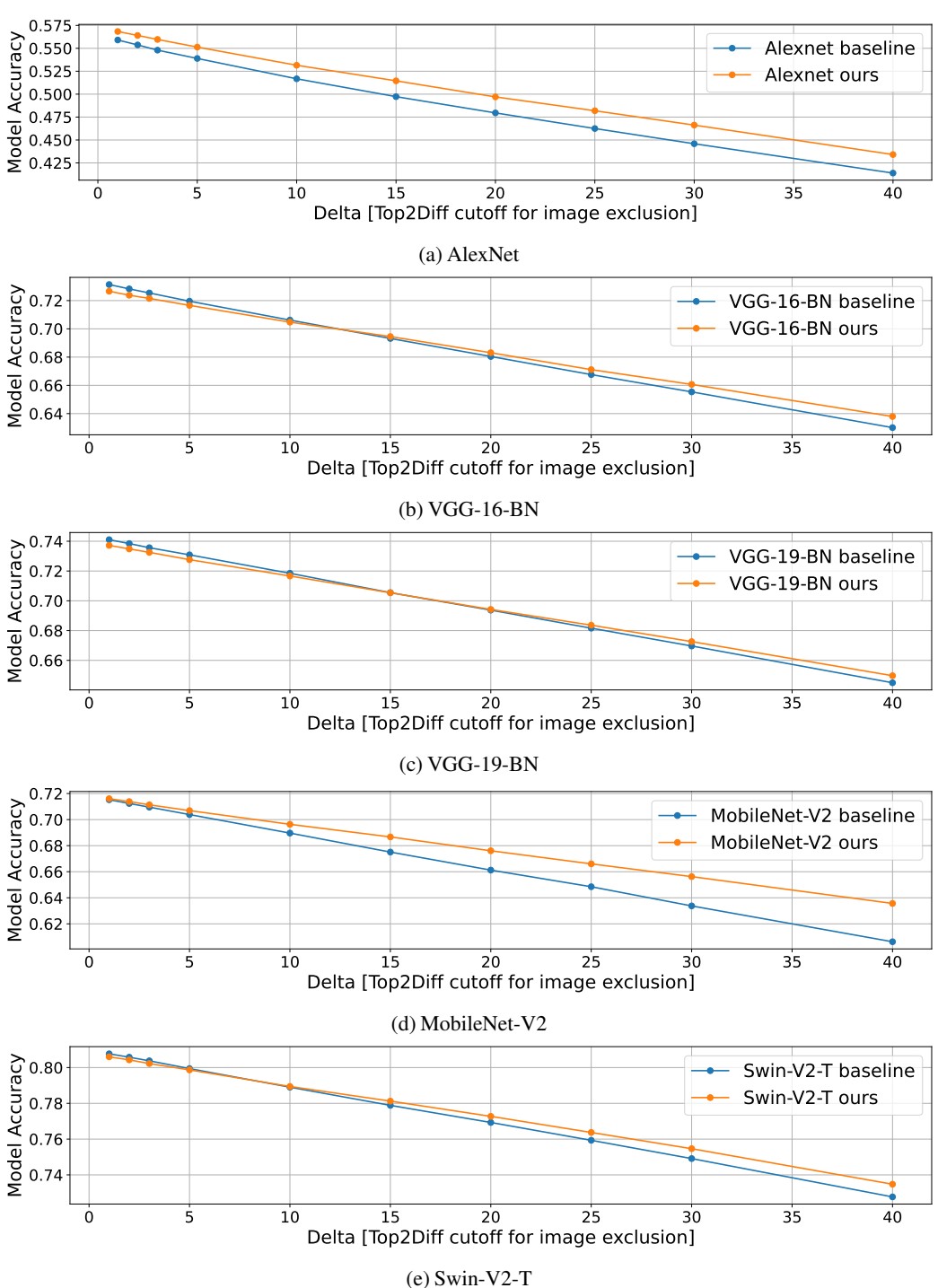

(a) AlexNet

(b) VGG-16-BN

(c) VGG-19-BN

(d) MobileNet-V2

(e) Swin-V2-T

Figure 11: **Model accuracy as a function of Top2Diff deltas.** This is an extension of Figure 4, for non-ResNet networks. We observe a similar trend, where our proposed technique's accuracy is more confident as you drop images with low Top2Diff, implying stronger confidence in classification.

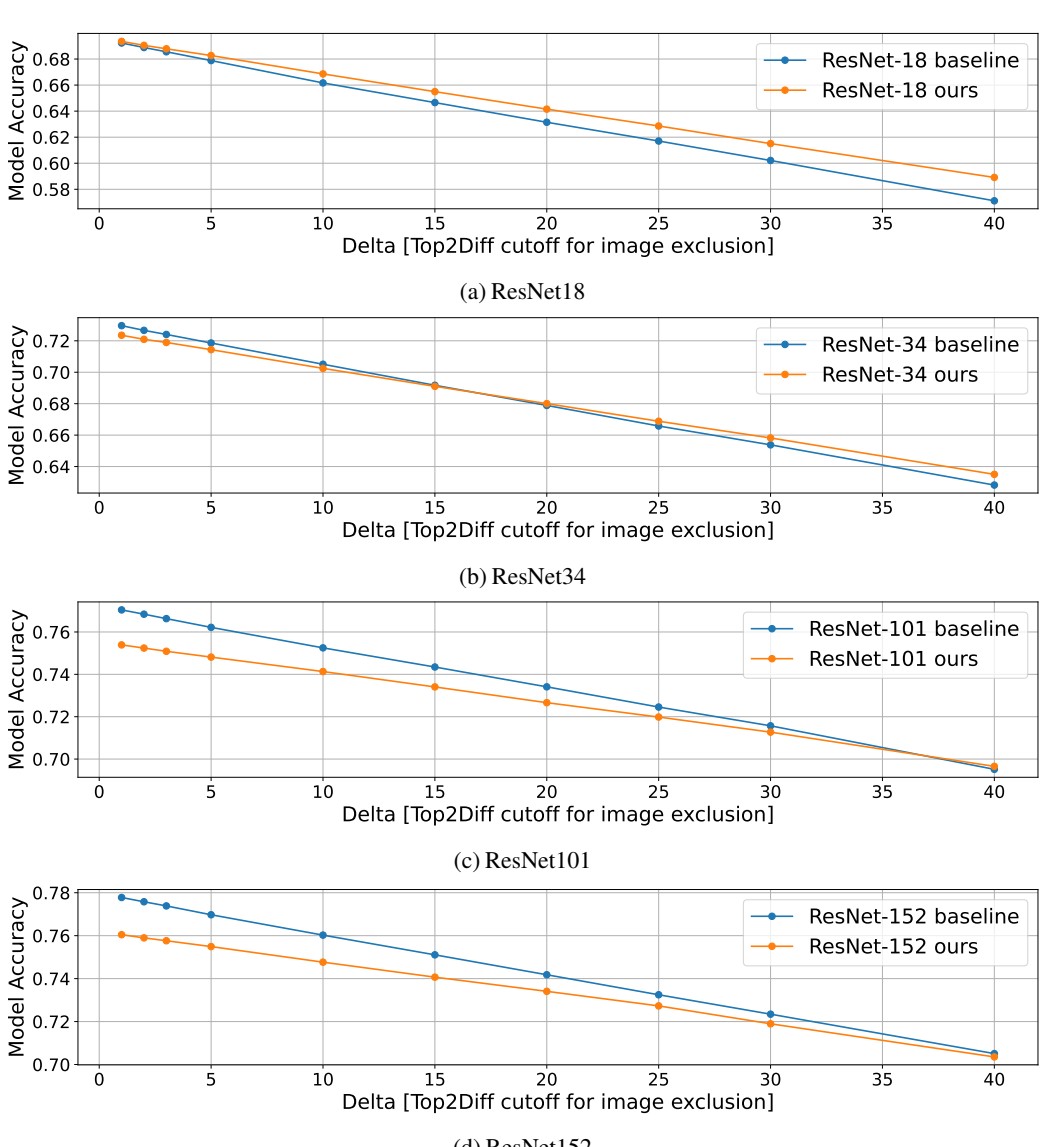

(a) ResNet18

(b) ResNet34

(c) ResNet101

(d) ResNet152

Figure 12: **Model accuracy as a function of Top2Diff deltas.** This is an extension of Figure 4, for ResNet-family networks. We observe a similar trend, where our proposed technique's accuracy is more confident as you drop images with low Top2Diff, implying stronger confidence in classification. The specific inflection point is network-dependent, but in all cases, our method's accuracy reduction is less sloped than the baseline.