# OpenReview forum: "Hardware Resilience Properties of Text-Guided Image Classifiers"
_NeurIPS.cc/2023/Conference — NeurIPS 2023 poster_

### Official Review · Reviewer_8rdZ · 2023-06-21

**Soundness:** 2 fair
**Presentation:** 2 fair
**Contribution:** 2 fair
**Rating:** 6
**Confidence:** 4

**Summary:**

The paper studies the hardware resiliency of image classifiers, that is misclassification rates under random-bit flipping. They show that initializing the last classification layer using CLIP embeddings can greatly improve hardware resiliency.

To obtain CLIP embeddings, for every class, GPT3 produces C different text prompts which are then averaged to produce a single embedding per class. In specific, the original classification layer of dim $B \times C$ is replaced by a latent layer of dim $B \times E$ and a projection layer of dim $E \times C$. The projection layer is initialized with embeddings from a CLIP text encoder. Results are shown on VGG and ResNet on 2 metrics (Top2Diff and $\Delta$ Loss which is difference in cross-entropy between the original prediction and misclassified prediction). Post hoc, the authors see that their proposed method has better saliency properties and lower last layer activations.

**Strengths:**

* The paper proposes a simple change with low overhead that improves hardware reliability rates.
* Since the clip embedding dimension is generally lower than classifier bottleneck dimension, a side effect is that the number of parameters are also reduced.
* Results are shown on a number of different convolutional architectures.

**Weaknesses:**

The positioning of the paper is a bit confusing. While finetuning pretrained clip models on ImageNet is not new in some sense, the paper seems to make the claim that for eg, in Figure 1 and Section 4.2 that this work proposes a new approach. The paper may be better positioned as an investigation of CLIP finetuning for hardware reliability. I would like to discuss with other reviewers about this aspect.

I have a few more questions about the clarity of some metrics and experiments used in this work, which may be useful to the broader computer vision community. Please see the questions section below.

**Questions:**

Metrics
--------
* It is a bit unclear to me what the "GoldenEye" benchmark actually is. The authors say "We use PyTorchFI [36] to perform the random bit flip, and we perform 4096 unique error injection experiments per layer, totaling more than 3.6 million experiments across all our models". Does the GoldenEye benchmark prescribe the exact way this error injection is done?
* I suggest that the authors report the classification flip rate as well in Table 1. The 14x number is interesting but is a bit unintuitive. Measuring how many decisions are flipped when the bit is flipped may be more intuitive.
* The Top2Diff metric is unclear. "Top2Diff metric is simply the difference in classification accuracy between the top (correct) class and the
second-highest class". I'm unsure what the classification accuracy of the second highest class means. Did you mean logits?

Experiments
---------------
* Major: What is the difference between RandomInit in Table 2 and the baseline in Table 1? If the projection layer is initialized randomly, then  does this not default to a baseline classifier?
*  Is layer 53 in Figure 3, the final classification layer meaning that the ResNet-50 baseline is highly confident? If yes, can the authors provide some intution on why small perturbations at the output layer changes its decision drastically?
* According to Figure 3, can an explicit additional L2 loss term at the outputs of the ResNet-50 baseline have the same effect as clip initialization?
* L305: The authors say that they inject 2000 random errors in weights across the network. Is the output saliency map averaged across these 2000 different maps?
* It might be nice to show some experiments on Vision Transformers, but this is more of a nice to have.

**Limitations:**

No negative societal impact.

---

> ### Author Rebuttal · Authors · 2023-08-09
>
> **q1:Positioning of the paper**
>
> **a1:** We would like to clarify that we don't exactly *finetune* CLIP models. We augment a standard, randomly initialized image model, with an additional projection head initialized with rich textual features. The rest of the model is randomly initialized as with any standard image classification training setup on the ImageNet dataset. Our major contribution is to the analysis that this simple additional projection + initialization to any image classification architecture can immediately improve the model resilience to hardware errors.
>
> We thank the reviewer for pointing this out. We will further clarify in the final version of the paper our major contribution focusing on the analysis of inherent resilience gains obtained by utilizing rich text features.
>
> **q2: GoldenEye clarifications**
>
> **a2:** Yes, we use the GoldenEye benchmark as provided in their repository for our evaluation. At a high level, GoldenEye is a wrapper around the PyTorchFI error injection framework plus providing us with an analysis of the results in terms of DeltaLoss resilience. To that end, we configure GoldenEye to perform 4096 unique error injection experiments per layer and use it to compare a baseline model with our proposed technique in terms of reliability. By using GoldenEye, the 3.6 million error injection experiments are performed under the hood to provide us with strong statistical guarantees on the model's resilience to hardware errors.
>
> **q3: Classification Bit Rate vs DeltaLoss**
>
> **a3:** The GoldenEye benchmark provides us with both the classification bit rate and the DeltaLoss metric as well. We report the DeltaLoss metric because the authors of [39] show that it converges asymptotically faster with fewer error injections compared to the classification bit rate, signifying that it is a stronger and more preferable metric for hardware resiliency analysis. In terms of the 14x number: it is the same whether reported through DeltaLoss or classical flip rate - the only difference is that the statistical guarantees are stronger via DeltaLoss at fewer error injection experiments (Figure 3 of [39] helps highlight this point).
>
> **q4: Top2Diff Metric is unclear**
>
> **a4:** The Top2Diff metric is computed for the *softmax*, not the logits. We apologize for the unclarity and will fix it.
>
> **q5: Difference between RandomInit in Table 2 and baseline in Table 1?**
>
> **a5:** The baseline in Table 1 is the default model as defined in the pytorch/torchvision library. The RandomInit model in Table 2 is our design with the additional projection, but no textual initialization to ablate whether the textual initialization helps or not.
>
> **q6: Figure 3, last layer Clarifications**
>
> **a6:** To clarify: Figure 3 does not refer to confidence. It shows empirical data for the maximum neuron value across the dataset, which we correlate with the single-bit flip error model as a reason for a high probability of a new (erroneous) value corrupting the output. We provide more detail in Appendix D of the Supplementary material.
>
> **q7:** Can an explicit L2 loss term be used at the output instead?
>
> **a7:** To see if adding an explicit L2 loss on the output helps, we run an experiment with this additional loss on the baseline ResNet-50 output. We show the results for this in the table below:
>
> |Backbone|Acc. Baseline|Acc. Last Layer L2 Norm|Improvement in Reliability (Last Layer)|Improvement in Reliability (Overall)| Improvement in Top2Diff|
> |:-:|:-:|:-:|:-:|:-:|:-:|
> |ResNet-50|75.64|75.13|-1.14x|-0.82x|-0.26%|
>
> We see that the explicit loss term reduces accuracy/resilience instead of improving it. We believe that the reason for this is that the training scheme already uses weight decay, which is also an implicit penalty on the model weights. Hence, the additional penalty does not really help much in terms of resilience.
>
> **q8: Figure 2 Clarifications**
>
> **a8:** For this experiment, there is only one saliency map generated by performing 2000 perturbations in the inference of an image. We chose this as a “large” number to show visual differences in the saliency between the two techniques. No averaging is needed. For error injection experiments, we perform a single bit flip in the entire network, and show that it can, in fact, alter the final classification.
>
> **q9: Additional Models**
>
> **a9:** We sincerely thank the reviewer for their suggestion to help improve this work. Included below, we have evaluated our proposed method on recent and state-of-the-art image classification models, to complement our CNN-based evaluation. Additionally, adding these models to our work strongly supports our original claims in the paper that our technique is general and can support any vision classification model type where we have the training recipe available.
>
> |Backbone|Acc. Baseline|Acc. Ours|Improvement in Reliability (Last Layer)|Improvement in Reliability (Overall)|Improvement in Top2Diff|
> |-|:-:|:-:|:-:|:-:|:-:|
> |FocalNet-T [1] (NeurIPS'22)|80.23|80.77|3.87x|2.61x|2.61%|
> |FocalNet-S [1] (NeurIPS'22)|82.01|82.52|4.73x|3.50x|3.10%|
> |Swin-V2-T [2] (CVPR'22)|80.97|80.02|1.65x|1.07x|2.85%|
> |Swin-V2-S [2] (CVPR'22)|82.71|82.86|3.51x|2.60x|3.04%|
> |MaxVit-T [3] (ECCV'22)|82.98|83.08|3.38x|2.63x|2.62%|
> |MobileNet-V2 [4] (CVPR'18)|71.87|71.83|3.92x|2.43x|5.36%|
>
> **References**
>
> [1] J. Yang, C. Li, X. Dai, and J. Gao, ‘Focal Modulation Networks’, in NeurIPS, 2022.
>
> [2] Z. Liu et al., ‘Swin Transformer V2: Scaling Up Capacity and Resolution’, in CVPR, 2022.
>
> [3] Z. Tu et al., ‘MaxViT: Multi-Axis Vision Transformer’, in ECCV, 2022.
>
> [4] M. Sandler, A. Howard, M. Zhu, A. Zhmoginov, and L.-C. Chen, ‘MobileNetV2: Inverted Residuals and Linear Bottlenecks’, in CVPR, 2018.
>
> [39] A. Mahmoud et. al., ‘Optimizing Selective Protection for CNN Resilience’, in ISSRE, 2021.

---

> > ### Comment · Reviewer_8rdZ · 2023-08-15
> > **Updated rating**
> >
> > I agree with the authors that this paper does not finetune clip models per-se but using clip textual features as the last layer is not entirely new and Section 4.2 implies that this is a contribution of this paper. This is misleading and can just be fixed by some minor editing.
> >
> > However, thanks to the authors for their additional experiments and the rebuttal. All my other concerns/questions are addressed and so I updated my rating to 6.

---

> > > ### Author Response · Authors · 2023-08-17
> > > **Thank you for the comments**
> > >
> > > We thank the reviewer for their comment. We value your suggestion and will update the final manuscript accordingly in Section 4.2 to clarify the contribution.
> > >
> > > Kind regards,
> > >
> > > Submission13786 Authors

---

### Official Review · Reviewer_eFF3 · 2023-07-07

**Soundness:** 3 good
**Presentation:** 4 excellent
**Contribution:** 3 good
**Rating:** 6
**Confidence:** 3

**Summary:**

This paper presents a software-based approach to enhance classifier resilience to hardware errors. It leverages GPT-3 to enhance text descriptions for target classes, followed by utilizing the CLIP text encoder to generate text embeddings. These embeddings initialize the classifier head, enabling it to learn robust representations by leveraging the generalization abilities of CLIP and GPT-3.

**Strengths:**

1. Using vision-language models to improve model resistance to hardware errors is an interesting idea.

2. The method is simple and effective. Experiments on ImageNet with VGG and ResNet show the model reliability increases significantly.

3. The paper is well-written and easy to follow. Figure 1 provides the data shapes at each step, making it easy to understand.

4. The experiment section includes rich ablation studies and intuitive explanations.

**Weaknesses:**

1. In Line 216, the delta isn't converted to a math symbol.

2. It seems that Top2Diff is not used correctly. The reference paper "Optimizing Selective Protection for CNN Resilience" defines Top2Diff as the difference between the top two class confidences. However, Line 231 assumes the top-1 class is a correct prediction. We don't know ground-truth labels during model deployment, so we can't tell whether the top-1 prediction is correct. But we still can compute Top2Diff, right? This is also related to the ablation study in Section 7.4. When removing images with certain Top2Diff,  do you remove only the images with correct predictions or all the images, no matter whether the predictions are correct?

3. The experiments don't use more recent backbones such as BEiT and Swin Transformer. ResNet and VGG are kind of out-of-dated.

**Questions:**

1. Do you train the classifier head, i.e., the projection layer, during training? If yes, do the head and the backbone use the same learning rate? If the head is updated significantly during training, would it lose the ability to guide the backbone learning representations?

2. In Table 1, why do some backbones (ResNet-18 and ResNet-34) have increased parameters, whereas others' parameters reduce?

3. For Figure 3, do you use the training set or validation set of ImageNet? Which set makes more sense for this purpose? Why use the maximum absolute value of one neuron rather than the average of absolute values of all neurons of one layer? The maximum value may favor more outliers.

**Limitations:**

It would be better to use some more recent vision backbones such as BEiT, DINO-v2, MAE, and Swin Transformer.

---

> ### Author Rebuttal · Authors · 2023-08-09
>
> **q1: Top2Diff Clarifications**
>
> **a1:** We thank the reviewer for highlighting this point. To clarify, we use the same exact procedure for hardware resiliency evaluation as the referenced paper [39]. It is true that we do not have a “ground truth” at runtime, but [39] claims that a strong indication of the probability an error occurs is the difference between the top 2 classes. During the evaluation, we focus only on the “correct” subset of images to perform error injections in, because it does not make sense to evaluate whether an error changed an incorrect class into a correct class. In deployment time, where we do not have the ground truth available, we would rely on our analysis of the model’s robustness, but would also need correction mechanisms. While our proposed technique does not propose a detection-and-recovery mechanism like [39], we introduce a technique that reduces the baseline probability of errors occurring in the first place, which would subsequently help lower the cost of techniques such as ILR and FLR proposed by [39].
>
> **q2: Evaluation on newer models**
>
> **a2:** We sincerely thank the reviewer for their suggestion to help improve this work. Included below, we have evaluated our proposed method on recent and state-of-the-art image classification models, to complement our CNN-based evaluation. Additionally, adding these models to our work strongly supports our original claims in the paper that our technique is general and can support any vision classification model type where we have the training recipe available. We find that the new models agree with our original assessments from the paper, that our technique not only helps improve the reliability of the last layer and the overall model but also has a negligible impact on the accuracy (and in some cases actually improves it), as was a primary goal in this research paper.
>
> |Backbone|Acc. Baseline|Acc. Ours|Improvement in Reliability (Last Layer)|Improvement in Reliability (Overall)|Improvement in Top2Diff|
> |-|:-:|:-:|:-:|:-:|:-:|
> |FocalNet-T [1] (NeurIPS'22)|80.23|80.77|3.87x|2.61x|2.61%|
> |FocalNet-S [1] (NeurIPS'22)|82.01|82.52|4.73x|3.50x|3.10%|
> |Swin-V2-T [2] (CVPR'22)|80.97|80.02|1.65x|1.07x|2.85%|
> |Swin-V2-S [2] (CVPR'22)|82.71|82.86|3.51x|2.60x|3.04%|
> |MaxVit-T [3] (ECCV'22)|82.98|83.08|3.38x|2.63x|2.62%|
> |MobileNet-V2 [4] (CVPR'18)|71.87|71.83|3.92x|2.43x|5.36%|
>
> **q3: Do you train the classifier head during training?**
>
> **a3:** Yes, we do train the classifier head during training, and yes, we use the same learning rate for both the head and the backbone to ensure we stay consistent with the overall training recipe.
>
> **q4: Why does ResNet-18,34 increase params, while ResNet50 decrease (Table 1)?**
>
> **a4:** This goes back to the DNN architecture design for different ResNet versions. As explained in lines 251-256, the second-to-last layer feeding into the projection layer impacts the change in the number of parameters for our model. If we take a look at Table 1 of the ResNet paper [5], we see that ResNet18 and ResNet34 end on a 3x3 512 layer, while the other ResNets end on a 1x1 2048 layer. More specifically:
>
> In case of the baseline Resnet18/34 last layer parameters are 512x1000 (num classes)  = 512000 \
> In case of our Resnet18/34 last layer parameters are 512x512 (embed size) + 512x1000 (proj layer)  = 262144 + 512000 \
> Hence params are increased by 0.26M
>
> In case of the baseline Resnet50/101/152 last layer parameters are 2048x1000 (num classes)  = 2048000 \
> In case of our Resnet18/34 last layer parameters are 2048x512 (embed size) + 512x1000 (proj layer)  = 1048576 + 512000 \
> Hence params are decreased by 0.49M
>
> **q5: Fig-3 elaboration:**
>
> **a5:** We use the validation set, as we are evaluating the network’s reliability outside the training regime. We conducted an analysis of the mean magnitude of neuron values for each layer, as per the reviewer's request. Complimenting Figure 3 in the main paper which uses absolute max value, we added a Figure in the rebuttal PDF showing the difference between the mean absolute value per layer for the baseline and our model. We see a similar trend between both figures, with the baseline model having a large value, especially in the final layer. This rationale is expounded upon in detail in Appendix D.
>
> **References:**
>
> [1] J. Yang, C. Li, X. Dai, and J. Gao, ‘Focal Modulation Networks’, in NeurIPS, 2022.
>
> [2] Z. Liu et al., ‘Swin Transformer V2: Scaling Up Capacity and Resolution’, in CVPR, 2022.
>
> [3] Z. Tu et al., ‘MaxViT: Multi-Axis Vision Transformer’, in ECCV, 2022.
>
> [4] M. Sandler, A. Howard, M. Zhu, A. Zhmoginov, and L.-C. Chen, ‘MobileNetV2: Inverted Residuals and Linear Bottlenecks’, in CVPR, 2018.
>
> [5] K. He, X. Zhang, S. Ren, and J. Sun, ‘Deep Residual Learning for Image Recognition’, in CVPR, 2016.
>
> [39] A. Mahmoud et. al., ‘Optimizing Selective Protection for CNN Resilience’, in ISSRE, 2021.

---

> > ### Author Response · Authors · 2023-08-18
> > **Request for Comments**
> >
> > Dear Reviewer eFF3,
> >
> > Thanks again for your effort in reviewing our paper and giving us a helpful chance to improve the paper's quality. We hope that our response can address your concerns.
> >
> > Considering that the discussion period will end on Aug 21st, we would like to know if you have any other questions about our paper, and we are glad to have a discussion with you in the following days. If our response has addressed your concerns, would you mind considering re-evaluating our work based on the updated information?
> >
> > Best regards,
> >
> > Submission13786 Authors

---

> > > ### Comment · Reviewer_eFF3 · 2023-08-20
> > > **About training the projection layer**
> > >
> > > Thank the authors for the response, which answers most of my questions. I still have one question regarding the training projection head initialized by the CLIP text embeddings. Have you tried the ablation study of freezing the head? I know there are one embedding projection layer and one classifier at the end of your networks. It makes sense to train the embedding projection layer since it's randomly initialized, but why do we need to tune the classifier initialized by text embeddings? Isn't it good to make the backbone+embedding projection layer produce features that are totally aligned with the text embeddings?

---

> > > > ### Author Response · Authors · 2023-08-21
> > > > **Regarding Freezing the Text Initialized Layer**
> > > >
> > > > Dear Reviewer eFF3,
> > > >
> > > > We did perform an experiment on a small-scale dataset (FOOD101) for freezing the last layer, the results of which we present below:
> > > >
> > > > |Dataset|Backbone|Acc. Baseline|Acc. Ours|Improvement in Reliability (Last Layer)|Improvement in Reliability (Overall)|Improvement in Top2Diff|
> > > > |:-:|:-:|:-:|:-:|:-:|:-:|:-:|
> > > > |FOOD101|ResNet-50 (Frozen Head)|83.13|82.47|1.38x|1.26x|1.97%|
> > > > |FOOD101|ResNet-50 (Fine-Tuned Head)|83.13|83.97|2.66x|2.15x|2.78%|
> > > >
> > > > We find that freezing the head still improves resilience but not as well as fine-tuning the last layer and also reduces accuracy slightly. We believe that the reason for this is that the last layer is not able to model the exact dataset distribution. When fine-tuned, we see that both the accuracy and resilience improve.
> > > >
> > > > We hope that the above answers the reviewer's question. Please let us know if there are any further questions.
> > > >
> > > > Best regards,
> > > >
> > > > Submission13786 Authors

---

### Official Review · Reviewer_DM9r · 2023-07-08

**Soundness:** 2 fair
**Presentation:** 3 good
**Contribution:** 2 fair
**Rating:** 6
**Confidence:** 4

**Summary:**

This paper provides a method to enhance the reliability of image classification models against hardware errors. For this purpose, the authors propose to combine textual and visual information to improve the reliability of neural networks by up to $14\times$, in comparison with traditional error detection and correction techniques. The authors verify their method on ImageNet classification with several models.

**Strengths:**

The proposed method to utilize texture information to enhance reliability of vision models is interesting and the author conducted some experiments on ImageNet classification with several models.

**Weaknesses:**

The proposed method requires textural information during inference, and thus the set of textural information collections would determine the final performance. For example, if the input image contains information outside those classes in ImageNet dataset, it is unclear how to use the proposed method during inference, except for retraining the models. Also, the authors only verify on early CNN architectures, not including efficient models or attention models, so the effectiveness of the proposed method might be questionable.

**Questions:**

Could the author explain how to apply the proposed method for cases when the image contains information outside the pretrained classes such as ImageNet in the paper?

**Limitations:**

The authors did not talk about limitations, and the question asked above can be one limitation of the proposed method.

---

> ### Author Rebuttal · Authors · 2023-08-09
>
> **q1: Does the proposed method require textual information during inference?**
>
> **a1:** No, the model only uses the textual information during *training*, to make it more resilient. Please note that in this paper we focus on a *closed-set* classification setting. We only use the text information to provide a rich initialization to the additional projection head. During inference, the model *only* uses the input image to perform its classification. As such, our technique produces an entirely drop-in solution where legacy models can be replaced with our new model, without breaking any previous abstractions of a system at inference time.
>
> **q2: Evaluation on newer models**
>
> **a2:** We sincerely thank the reviewer for their suggestion to help improve this work. Included below, we have evaluated our proposed method on recent and state-of-the-art image classification models, to complement our CNN-based evaluation. Additionally, adding these models to our work strongly supports our original claims in the paper that our technique is general and can support any vision classification model type where we have the training recipe available. We find that the new models agree with our original assessments from the paper, that our technique not only helps improve the reliability of the last layer and the overall model but also has a negligible impact on the accuracy (and in some cases actually improves it), as was a primary goal in this research paper.
>
> |Backbone|Acc. Baseline|Acc. Ours|Improvement in Reliability (Last Layer)|Improvement in Reliability (Overall)|Improvement in Top2Diff|
> |-|:-:|:-:|:-:|:-:|:-:|
> |FocalNet-T [1] (NeurIPS'22)|80.23|80.77|3.87x|2.61x|2.61%|
> |FocalNet-S [1] (NeurIPS'22)|82.01|82.52|4.73x|3.50x|3.10%|
> |Swin-V2-T [2] (CVPR'22)|80.97|80.02|1.65x|1.07x|2.85%|
> |Swin-V2-S [2] (CVPR'22)|82.71|82.86|3.51x|2.60x|3.04%|
> |MaxVit-T [3] (ECCV'22)|82.98|83.08|3.38x|2.63x|2.62%|
> |MobileNet-V2 [4] (CVPR'18)|71.87|71.83|3.92x|2.43x|5.36%|
>
> **q3: How does the proposed method work when images contain information outside pre-trained classes?**
>
> **a3:** To clarify, we do not work in an *open-set/zero-shot classification* setting. The proposed technique aims at improving the resilience of image classification models in the *closed-set classification* supervised setting. We only use the textual features to provide a rich initialization to the new projection head added to the network which improves the overall resilience of the network. Afterward, we follow a standard supervised training and testing setting.
>
> **References:**
>
> [1] J. Yang, C. Li, X. Dai, and J. Gao, ‘Focal Modulation Networks’, in NeurIPS, 2022.
>
> [2] Z. Liu et al., ‘Swin Transformer V2: Scaling Up Capacity and Resolution’, in CVPR, 2022.
>
> [3] Z. Tu et al., ‘MaxViT: Multi-Axis Vision Transformer’, in ECCV, 2022.
>
> [4] M. Sandler, A. Howard, M. Zhu, A. Zhmoginov, and L.-C. Chen, ‘MobileNetV2: Inverted Residuals and Linear Bottlenecks’, in CVPR, 2018.

---

> > ### Author Response · Authors · 2023-08-18
> > **Request for Comments**
> >
> > Dear Reviewer DM9r,
> >
> > Thanks again for your effort in reviewing our paper and giving us a helpful chance to improve the paper's quality. We hope that our response can address your concerns.
> >
> > Considering that the discussion period will end on Aug 21st, we would like to know if you have any other questions about our paper, and we are glad to have a discussion with you in the following days. If our response has addressed your concerns, would you mind considering re-evaluating our work based on the updated information?
> >
> > Best regards,
> >
> > Submission13786 Authors

---

> > > ### Comment · Reviewer_DM9r · 2023-08-21
> > > **Updated rating**
> > >
> > > Dear authors,
> > >
> > > Thanks for your response. My concerns are dealt with properly after the discussion and I have updated my rate.
> > >
> > > Thanks.
> > >
> > > Best

---

> > > > ### Author Response · Authors · 2023-08-21
> > > > **Thank you**
> > > >
> > > > We would like to express our gratitude to the reviewer for their comment. We appreciate your effort in reviewing our paper and rebuttal.
> > > >
> > > > Kind regards,
> > > >
> > > > Submission13786 Authors

---

### Official Review · Reviewer_4rtm · 2023-07-15

**Soundness:** 2 fair
**Presentation:** 3 good
**Contribution:** 2 fair
**Rating:** 6
**Confidence:** 2

**Summary:**

This paper studies how CLIP can be used to mitigate the effect of hardware failures on image classification models. The authors do this by incorporating embeddings from the clip text encoder of class based inputs (queried through GPT) to initialize the classification layer. The authors evaluate this technique by comparing how it affects accuracy on the downstream task, as well as some hardware reliability evaluation.

**Strengths:**

- Presentation is clear and explains why this is an important problem to work on for someone not familiar with the area

- Method is simple and generalizes to standard image classification models

- Evaluation metrics make sense (maintaining accuracy, investigating reliability)

- Ablations are interesting and cover initial questions from reviewing the results section

**Weaknesses:**

- Since you are using the CLIP text encoder, I wish there was more exploration in that space. I think a natural question that is in-scope is how robust are CLIP models both for zeroshot and fine-tuning to hardware failure.
- Evaluation is limited to just ImageNet
- Confidence (top2diff) might not be an informative metric and doesn't necessarily make a better model - perhaps something related to calibration could be helpful

**Questions:**

See weaknesses - mostly concerned about evaluation and comparison

---

> ### Author Rebuttal · Authors · 2023-08-09
>
> **q1: How robust are CLIP models to hardware failure?**
>
> **a1:** We perform error injection on the CLIP model in the zero-shot setting below and compare it against the baseline (a standard model trained on ImageNet in a supervised manner). We use the resnet50 backbone version of CLIP.
>
> |Backbone|Acc. Baseline|Acc. CLIP ZeroShot|Improvement in Reliability (Last Layer)|Improvement in Reliability (Overall)|Improvement in Top2Diff|
> |-|:-:|:-:|:-:|:-:|:-:|
> |ResNet-50|75.64|58.18|-5.38x|-3.71x|-2.85%|
>
> We see that zero-shot CLIP has much lower resilience than the supervised trained model on ImageNet. We believe that two reasons contribute to this. Firstly, the CLIP has not seen the exact dataset distribution like the supervised baseline. And Secondly, the CLIP model uses both a vision and text encoder during inference. So it has a larger number of parameters that are susceptible to bit flips that can cause a misclassification.
>
> **q2: Beyond ImageNet evaluation.**
>
> **a2:** We thank the reviewer for highlighting this point and have included additional evaluation spanning multiple datasets (CIFAR10, CIFAR100, Food101, and STL10) for two networks: ResNet-50 [1] and FocalNet-T [2]. Our results, shown below in the table, validate that our technique is general and can work across an array of model types and datasets. Furthermore, as we have shown in the paper, we did not have to modify any hyperparameters in the process, suggesting the ease of our technique as well as the increased benefit from a reliability point of view. Additionally, adding these new datasets further support our claims made in Section 7.4 of the paper, that our technique has negligible impact on model training accuracy, whilst still providing us with a large upside in resilience.
>
> |Dataset|Backbone|Acc. Baseline|Acc. Ours|Improvement in Reliability (Last Layer)|Improvement in Reliability (Overall)|Improvement in Top2Diff|
> |-|:-:|:-:|:-:|:-:|:-:|:-:|
> |CIFAR10|ResNet-50|95.07|95.29|2.04x|1.71x|6.70%|
> |CIFAR10|FocalNet-T|94.76|94.94|2.47x|1.30x|3.58%|
> |CIFAR100|ResNet-50|78.23|78.53|2.19x|1.65x|3.69%|
> |CIFAR100|FocalNet-T|77.06|79.21|3.21x|1.58x|2.90%|
> |FOOD101|ResNet-50|83.13|83.97|2.66x|2.15x|2.78%|
> |FOOD101|FocalNet-T|85.64|85.91|3.28x|2.85x|1.70%|
> |STL10|ResNet-50|47.73|52.68|2.10x|1.91x|2.45%|
> |STL10|FocalNet-T|62.74|63.78|2.23x|1.72x|1.96%|
>
> **q3: Confidence (Top2Diff) as an informative metric for improving a model?**
>
> **a3:** To clarify: Top2Diff is an *analysis* metric, and it is measured post-training (as the authors of [39] suggest), as opposed to an optimization/cost metric *for* training.  The metric itself points to the propensity that an error could sufficiently cause a misclassification at the output, and as such, increasing the gap between the first and second-class confidence, a hardware error manifestation is less likely to corrupt the output. Our insight that Top2Diff can be used to understand the relationship between resilience and accuracy is strongly supported by our original experiments in Section 7.4, plus the addition of the new datasets which we thank the reviewer for suggesting we include in our paper.
>
> Additionally, with regard to model calibration, we compare and contrast Model Calibration with the Top2Diff Metric in the table below:
>
> |Aspect|Model Calibration|Top2Diff|
> |-|-|-|
> |Nature|Technique to adjust predicted probabilities to match true probabilities.|Analysis metric quantifying the difference between the top two predicted class probabilities.|
> |Purpose|Ensure well-calibrated predicted probabilities for accurate confidence estimates.|Indicate susceptibility of predictions to errors and assess the resilience-accuracy relationship.|
> |Timing|Performed post-training to refine predicted probabilities.|Measured post-training to analyze model behavior.|
> |Impact|Refines predicted probabilities to align with true probabilities.|Quantifies potential impact of errors on predictions and resilience-accuracy trade-off.|
>
> While both calibration and Top2Diff address the reliability and accuracy of machine learning models, calibration focuses on refining the probability estimates to align with reality, while Top2Diff serves as a post-training metric to measure the potential impact of errors on the model's predictions and understand its resilience-accuracy trade-off. We would like to clarify that while calibration could potentially be used as an inference-based technique to improve model resilience, our goal as outlined in Lines 43-47 was to introduce a low-entry and entirely training-based routine to reduce the inherent and underlying vulnerability of a model significantly, after which many inference-side techniques could potentially be appended for even stronger resilience (including calibration, as proposed by this reviewer, or other selective protection mechanisms as outlined in Lines 39-43).
>
> **References:**
>
> [1] K. He, X. Zhang, S. Ren, and J. Sun, ‘Deep Residual Learning for Image Recognition’, in CVPR, 2016.
>
> [2] J. Yang, C. Li, X. Dai, and J. Gao, ‘Focal Modulation Networks’, in NeurIPS, 2022.
>
> [39] A. Mahmoud et. al., ‘Optimizing Selective Protection for CNN Resilience’, in ISSRE, 2021.

---

> > ### Author Response · Authors · 2023-08-18
> > **Request for Comments**
> >
> > Dear Reviewer 4rtm,
> >
> > Thanks again for your effort in reviewing our paper and giving us a helpful chance to improve the paper's quality. We hope that our response can address your concerns.
> >
> > Considering that the discussion period will end on Aug 21st, we would like to know if you have any other questions about our paper, and we are glad to have a discussion with you in the following days. If our response has addressed your concerns, would you mind considering re-evaluating our work based on the updated information?
> >
> > Best regards,
> >
> > Submission13786 Authors

---

> ### Comment · Reviewer_4rtm · 2023-08-19
>
> Thank you for your thorough response. I have updated my score to account for the authors' response.

---

> > ### Author Response · Authors · 2023-08-21
> > **Thank you**
> >
> > We would like to express our gratitude to the reviewer for their comment. We appreciate your effort in reviewing our paper and rebuttal.
> >
> > Kind regards,
> >
> > Submission13786 Authors

---

### Author Rebuttal · Authors · 2023-08-09

We thank the reviewers for all their feedback and comments. We respond to common themes across all reviewers in this global rebuttal, and then answer further specific questions per reviewer individually.

**Q1. Vision model evaluation for hardware resilience, and the use of CLIP-text features**

One of the fundamental points of this work is that we use textual features to enhance the resilience of *closed-set image classification models*. One common misconception we gathered was about the use of textual information during inference. Rather, we operate entirely within a closed-set regime in this paper and only use the textual information from CLIP only to initialize the projection layer during the *training* of our model. Thus, at deployment time, there is no need for explicit textual information, as that has already been incorporated into the model via our technique of pre-training the final layer of the model. To that end, our proposed model can directly replace any classical model and would have the same pros and cons (in the context of in-distribution and out-of-distribution performance), but operate at a higher level of resilience to single-bit perturbations and hardware errors.

In summary, our approach involves refraining from fine-tuning CLIP models directly. Instead, we enhance a conventional image model, which starts as a randomly initialized framework, by incorporating an extra projection head initialized with textually enriched features. The remaining segments of the model are also subject to random initialization, following the conventional image classification training process using the ImageNet dataset. Notably, our primary contribution lies in demonstrating that this uncomplicated augmentation through an extra projection and initialization step, when applied to any image classification architecture, can swiftly enhance the model's ability to withstand hardware errors.

**Q2. Top2Diff Discussion**

We credit the authors of [39] for the introduction of the Top2Diff metric in the context of hardware reliability and would like to further clarify its use here. Top2Diff is a measurement between the top softmax value and the 2nd highest softmax value. In Section IV-B of [39], the authors show that this metric is good for hardware reliability, intuitively because it indicates a higher threshold that an error needs to overcome for the output to change its classification. For example, a Truck with 52% confidence and Bird with 48% confidence indicates a Top2Diff of 4%, while a 90% Truck to 10% Bird indicates an 80% Top2Diff. The idea is that “overcoming” a 4% difference is a much lower threshold than an 80% difference, implying that a single bit flip during the first Truck example is more likely to produce an output misclassification.

To further clarify, we do not use Top2Diff during training at all, and simply measure it as a proxy to understand the reliability of a model to hardware errors. The error injection experiments produced can be considered the primary metric for resilience (which is measured by DeltaLoss), and Top2Diff is helpful to gather insight into the model only.

**Q3. Additional Results**

We included additional results as requested by multiple reviewers, including:
- Adding more models including modern transformers (FocalNet, Swin-V2, MaxVit, and MobileNet-V2)
- Adding additional datasets (CIFAR10/100, STL10, and Food101)
- Including an additional metric (absolute mean) to complement our original metric (absolute max) in Figure 3 as requested by one of the reviewers. This is presented in the rebuttal PDF file.

|Backbone|Acc. Baseline|Acc. Ours|Improvement in Reliability (Last Layer)|Improvement in Reliability (Overall)|Improvement in Top2Diff|
|-|:-:|:-:|:-:|:-:|:-:|
|FocalNet-T [1] (NeurIPS'22)|80.23|80.77|3.87x|2.61x|2.61%|
|FocalNet-S [1] (NeurIPS'22)|82.01|82.52|4.73x|3.50x|3.10%|
|Swin-V2-T [2] (CVPR'22)|80.97|80.02|1.65x|1.07x|2.85%|
|Swin-V2-S [2] (CVPR'22)|82.71|82.86|3.51x|2.60x|3.04%|
|MaxVit-T [3] (ECCV'22)|82.98|83.08|3.38x|2.63x|2.62%|
|MobileNet-V2 [4] (CVPR'18)|71.87|71.83|3.92x|2.43x|5.36%|


|Dataset|Backbone|Acc. Baseline|Acc. Ours|Improvement in Reliability (Last Layer)|Improvement in Reliability (Overall)|Improvement in Top2Diff|
|-|:-:|:-:|:-:|:-:|:-:|:-:|
|CIFAR10|ResNet-50|95.07|95.29|2.04x|1.71x|6.70%|
|CIFAR10|FocalNet-T|94.76|94.94|2.47x|1.30x|3.58%|
|CIFAR100|ResNet-50|78.23|78.53|2.19x|1.65x|3.69%|
|CIFAR100|FocalNet-T|77.06|79.21|3.21x|1.58x|2.90%|
|FOOD101|ResNet-50|83.13|83.97|2.66x|2.15x|2.78%|
|FOOD101|FocalNet-T|85.64|85.91|3.28x|2.85x|1.70%|
|STL10|ResNet-50|47.73|52.68|2.10x|1.91x|2.45%|
|STL10|FocalNet-T|62.74|63.78|2.23x|1.72x|1.96%|

---

### Decision · Program_Chairs · 2023-09-21

**Decision:**

Accept (poster)

**Comment:**

This paper introduces a method to enhance the reliability of image classification models against hardware errors. They specifically study how they can mitigate random bit flips in image classification models. The reviewers agree this is an important and interesting problem to study. The reviewers agree that this solution seems simple and effective. There is an added bonus that this model has less flops. The authors do a good job setting up the problem of reliability and provide + motivate a metric to study reliability.